# A qualitative study of perceived barriers and facilitators to point-of-care ultrasound use among Veterans Affairs Emergency Department providers

**Rebecca G. Theophanous**[1,2]*, **Anna Tupetz**[1,3], **Luna Ragsdale**[1,2], **Padmaja Krishnan**[4], **Raelynn Vigue**[1], **Carson Herman**[1], **Jaran White**[1], **Catherine A. Staton**[1,3], **Stephanie A. Eucker**[1]

1 Department of Emergency Medicine, Duke University School of Medicine, Durham, NC, United States of America, 2 Durham Veterans Affairs Healthcare System, Durham, NC, United States of America, 3 Global Health Institute, Duke University, Durham, NC, United States of America, 4 Campbell University School of Medicine, Lillington, NC, United States of America

* rebecca.theophanous@duke.edu

**Data Availability Statement:** All relevant data are within the manuscript and its Supporting Information files.

## Abstract

Consistent point-of-care ultrasound (POCUS) use and retention is difficult to achieve, with prior studies citing a lack of provider training, credentialed ultrasound users, and image review as contributing factors. We aimed to assess user feedback on a POCUS implementation intervention by identifying and characterizing the perceived barriers and facilitators at a single Veterans Affairs (VA) hospital using the consolidated framework for implementation research (CFIR). We implemented a co-designed multifaceted training intervention at a VA emergency department (ED) to enhance POCUS usability and sustainability from November 2021-October 2022. We performed semi-structured interviews with 13 attending physicians and 1 Advanced Practice Provider (average of 15 years of clinical practice) in August-October 2022. Interviews were audio-recorded, transcribed, and double-coded using inductive content analysis and mapped to the CFIR, using deductive coding strategies. Through inductive analysis, five major themes emerged: 1) POCUS workflow convenience and efficiency, 2) ED environment and resources, 3) perceptions of high clinical utility of POCUS, 4) perceptions of high educational utility of POCUS, and 5) peer influences, feedback, and teaching. Within these major themes, POCUS facilitator subthemes include: machine availability, use in resident teaching, use in ED procedures, hands-on group training, colleagues' contagiousness and enthusiasm, and support from ultrasound faculty, ED, and hospital leadership. POCUS barrier subthemes were: time constraints, alternative radiology imaging availability, cumbersome steps for image acquisition and documentation/storage, and limited POCUS knowledge and skills comfort. Additional needs identified through CFIR mapping (archiving software, image review process, and faculty credentialing), require development locally to strengthen provider skills and reduce duplicated radiology studies. Our model is a reproducible clinical tool to evaluate barriers and facilitators to POCUS program implementation at any site. Future work should tailor POCUS education to individuals,

**Funding:** RGT received a Society of Academic Emergency Medicine Foundation/Academic Emergency Ultrasound Section grant #AG 2020-0000000099. https://www.saem.org/about-saem/academies-interest-groups-affiliates2/saem-foundation/apply-for-a-grant/academy-grants/aeus-research-grant. The funders had no role in study design, data collection and analysis, decision to publish, or preparation of the manuscript.

use momentum from positive peer feedback including "ED clinical champions", and integrate ED/hospital leadership support for program sustainability.

## 1 Introduction

Point-of-care ultrasound (POCUS) is an important bedside tool used by emergency providers in medical decision making and critical procedural skills such as intravenous access.[1] Over the past 20 years, POCUS has become integral to emergency care due to its low cost, lack of ionizing radiation, ease of use, and portability [1,2]. POCUS has also been integrated as a core clinical competency in residency programs including emergency medicine (EM), internal medicine (IM), and critical care [1–3]. However, problems persist with heterogeneity in POCUS utilization and skill retention through sustained frequency of POCUS use nationwide, at both community and academic emergency departments (ED) [2–4].

The Veterans Affairs (VA) is the largest integrated healthcare system in the United States, treating 9 million veterans at 171 VA Medical Centers and 1298 healthcare facilities nationwide, with over 2 million ED and urgent care visits annually [5,6]. Despite progress in ultrasound equipment upgrades and availability, a recent study of the VA healthcare system identified large gaps in POCUS knowledge and utilization at many VA facilities, with only 53% (61/115) of sites using POCUS clinically for patient care [7,8]. No cohesive POCUS guidelines exist in the VA healthcare system, and individual VA site POCUS use is highly variable based on the active practice and skills of local clinicians [7,8]. Furthermore, a high desire for POCUS training was identified in prior studies, but barriers still exist to successful program implementation and sustainability.

Qualitative methods enable a deeper dive into the impact of barriers, elucidate additional barriers, and explore how to address and overcome these barriers to better facilitate successful interventions for POCUS use and knowledge retention. Potential barriers to sustained POCUS use identified at both VA and non-VA sites from survey-based studies include a lack of EM physician training, lack of credentialed ultrasound users, and lack of quality assurance image review programs [9–13]. Similar barriers were identified for both attending physicians and resident trainees, highlighting the need for standardized curricula, documentation procedures, and POCUS workflows [14,15]. To address this gap, we implemented a co-designed multifaceted intervention at a single VA ED to achieve POCUS sustainability and knowledge retention [16].

This complex process required more in-depth evaluation of the context, intervention, process, and individuals to understand how to overcome unique multi-dimensional barriers. We co-designed POCUS training based on a participant baseline needs assessment survey, with detailed description in a separate paper [16–18].

Our aim for this study was to perform a qualitative analysis of the perceived barriers and facilitators to POCUS use by clinical providers (attending physicians and advanced practice providers (APPs)) at a single VA ED using the consolidated framework for implementation research (CFIR), which is user-informed by collecting data from the individual participants and key stakeholders [19]. The qualitative analysis presented here reports on the participant experience with training, implementation, clinical POCUS use, and knowledge retention after undergoing POCUS training; and it serves as a reproducible model to inform future implementation at other sites.

## 2 Methods

### 2.1 Study design

We performed a qualitative assessment of our co-designed intervention regarding the perceptions of VA ED providers on barriers and facilitators to POCUS use in clinical care. We performed CFIR framework-driven semi-structured interviews in August-October 2022. Our study was determined to be exempt by the VA Institutional Review Board (1631300–4). We report our findings according to the Consolidated Criteria for Reporting Qualitative Research [20].

**Setting.** Data was collected at a single site Veterans Affairs ED (DVAHCS). All providers were employed at the DVAHCS, a 251-bed tertiary care referral, teaching, and research facility affiliated with a university. The DVAHCS ED evaluates and treats approximately 26,000 adult patients annually, with a 16.3% admission rate. From November 2021-October 2022, the most common presenting diagnoses included chest pain, abdominal pain, shortness of breath, dizziness, and syncope. Our study was deemed exempt by the VA Institutional Review Board, and consent was waived by the IRB (1631300–4).

### 2.2 Research team and reflexivity

The PI RT designed the research study (EM clinician and emergency ultrasound director, MD, MHSc) with mentorship by AT (post-doctoral researcher with experience in implementation science and qualitative research training), SE (EM clinician, MD, PhD, Assistant Director of Emergency Research), and CS (EM clinician, MD, MScGH, Director of the GEMINI (Global EM Innovation & Implementation) Research Center and the EM Vice Chair of Research Strategy & Faculty Development). LR (EM clinician, MD, MPH and VA ED Chief) was involved in the co-design process, collaboration with VA staff and ED leadership support of POCUS endeavors, and monthly feedback meetings. PK conducted the semi-structured interviews (second-year medical student, DO candidate, BSc, MSPH with prior qualitative interview experience for her master's degree). RV, JW, and CH are undergraduate or medical students trained in qualitative research methods who performed team-based interview coding with the study PI. AG and MK performed statistical analysis for quantitative survey data and ED metrics. All researchers are females except JW and AG who are male.

### 2.3 Selection of participants

All attendings and advanced practice providers (APPs) (n = 39) working in the site's VA ED at the time of the study period were invited to participate. We recruited primary ED providers (24 attendings and 1 APP), rather than moonlighting fellows (n = 14) who may rotate through the ED on a monthly or yearly basis. Nursing and intermediate care technicians (ICT or advanced health technicians with intensive specialized military medical training) were not included given that their ultrasound use and training purposes differ from those of the attending/APP group. We used a convenience sampling strategy and recruited participants via word of mouth, ED provider list emails, and announcements at monthly VA ED provider meetings from April 1, 2022 to October 31, 2022.

**Co-design of the intervention.** Details of the co-design and POCUS training intervention are described in detail elsewhere. Briefly, VA ED leadership (ED Chief, deputy director, committee and educational leads) and ultrasound faculty met and developed a co-designed educational curriculum, plus optimal processes for POCUS image acquisition, documentation, archiving, and quality assurance review. We convened monthly to analyze and provide direct feedback on further improving our POCUS program [16].

For the educational interventions, ED providers voluntarily participated in a hands-on large-group course in May 2022 and monthly reinforcement sessions taught by ultrasound faculty, based on a validated ultrasound educational format [21–24].

## 2.4 Data collection

We developed semi-structured interview questions to characterize barriers and facilitators during and after the implementation of a POCUS program. Interviews were topic guided with open-ended questions to explore individual provider perspectives and allow for new ideas and informed by literature evaluating barriers and facilitators to POCUS use in other ED settings [12,13,15,25]. We structured questions based on the consolidated framework for implementation research (CFIR) to ensure data capture from all stakeholders including the individuals, group, inner setting (ED providers and ED leadership), and outer setting (VA hospital leadership, national policies) (Fig 1) [19,26]. Using the Interview Protocol Refinement (IPR) framework, questions were refined based on a pilot interview with an ED physician from another local ED who was not eligible for the study to ensure clarity of questions and adequate content capture [27]. The interview guide can be found in the supplement (S1 Appendix).

In-person interviews were held in a private room at the VA ED and audio recorded with an encrypted voice recording device, while virtual interviews took place in a private space using a VA-secured version of Cisco WebEx or Microsoft Teams. Verbal consent was obtained by reading aloud a written statement describing voluntary study participation at the beginning of each audio recorded interview and documented using a data collection checklist. One female trained research assistant (PK - MSPH, D.O. candidate) with prior qualitative interview experience and no relationship to the participants conducted all 14 semi-structured interviews between August-October 2022. She was introduced to the participants by the study PI. This was after the co-designed large-group training session but during monthly reinforcement teaching sessions, to assess user feedback. Interviews were approximately 30 minutes, and all 14 interviews were transcribed by RT and PK into Microsoft Word using a naturalistic approach, then deidentified and uploaded into NVivo [19,28]. Data saturation was reached when there were no new emergent themes after three consecutive interviews. Interviews were de-identified and a key created by the study PI prior to research team coding and data analysis.

## 2.5 Data analysis

Qualitative analysis was performed from September 15, 2022 to January 31, 2023 using grounded theory components, which incorporates qualitative content analysis to derive coding categories directly from raw data [29]. Thematic content analysis was chosen to first identify descriptive codes (subthemes) and then progress towards themes (inductive coding) and lastly framework application (deductive coding) [30]. We first identified emerging themes and subthemes after reading each interview transcript. Emergent subthemes were then organized into themes with conceptual similarities. We then performed deductive content analysis by applying the identified themes to the CFIR framework as a structured approach to analyze the data and test concepts in the clinical context (Fig 1) [14]. For thematic content analysis, two researchers (RT and PK) developed the initial codebook through conventional content analysis of three interview transcripts by independently identifying themes and subthemes based on conceptual similarities. These were discussed and a selective coding approach used to generate the codebook. The codebook was iteratively adapted based on new emergent themes and discussions with the full study team. A team of four researchers (RT, RV, JW, CH) completed subsequent coding with a common codebook using NVivo 12 software, with coaching from an implementation science and qualitative research trained researcher (AT). Each interview was

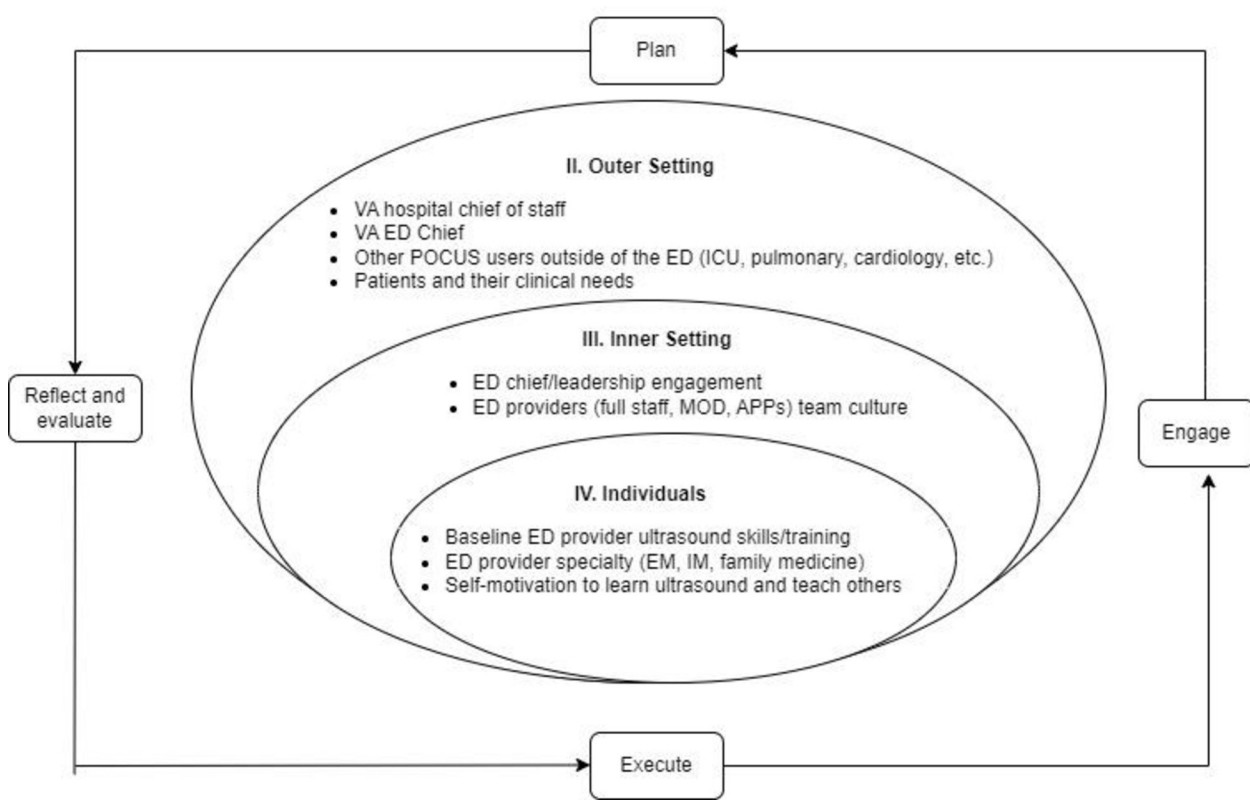

I. Intervention characteristics (e.g. Features of a POCUS program at a single site Veterans Affairs Emergency Department)

**Fig 1. Consolidated framework for implementation research framework.**

double-coded independently by two team members to cross-validate results, with differences reconciled by consensus, and all interviews were combined into a final codebook after discussions between the team members (RT, RV, JW, CH). In cases where consensus was not reached, differences were resolved by the PI (RT). RT created descriptive summaries for each emerging theme and subtheme, summarizing the findings in an ongoing fashion. The descriptive summaries served as a basis for discussion and feedback to the entire research team. All coding team members reviewed the emergent themes and preliminary results for content validity and to ensure interpretation accuracy [28].

## 3. Results

### 3.1 Participant characteristics

Of the 39 total ED providers at the study site, 24 were primary ED attendings, 1 was an APP, and 14 were moonlighting physicians. Of these, 14 primary ED providers consented to participate in semi-structured interviews. No participants dropped out of the study. Participant ages ranged from 30–65 years old. 8 participants identified as female and 6 as male. 13 were attending physicians (11 EM, 2 IM), and 1 APP. For additional training, participants completed geriatric EM (n = 1), Health Services Research (n = 1), and Emergency Medical Services (n = 1) fellowships. Professional experience ranged from 0–10 years (n = 4), 11–20 years (n = 6), and >20 years (n = 4). About 40% of ED providers self-reported subscribing to the self-directed

online learning database. Table 1 describes the participant characteristics. The study timeline is presented in Fig 2.

## 3.2 Results

We identified five major themes regarding ED provider perceived barriers and facilitators to POCUS use: 1) POCUS workflow convenience and efficiency, 2) ED environment and available resources (space and place), 3) perceptions of high clinical utility of POCUS in the ED (medical decision making), 4) perceptions of high educational utility of POCUS, and 5) peer influences, feedback, and teaching.

Within each theme, there were underlying subthemes corresponding to (1) **facilitators** and (2) **barriers** based on the CFIR construct to which the theme relates. These subthemes describe perceived facilitators and barriers to individual POCUS knowledge and skills, POCUS image documentation and archiving processes, ultrasound equipment management and upkeep, faculty credentialing, and collaboration both within and outside of the ED for a comprehensive clinical POCUS program (Table 2).

**Table 1. Participant characteristics of Veterans Affairs Emergency Department providers from semi-structured interviews.**

| Participant ID | Sex | Specialty | Leadership/role | Age range | Years working in EM | Prior ultrasound training | Number of personally performed POCUS in medical school? |
|---|---|---|---|---|---|---|---|
| VAR001 | female | EM | none (VA opioid committee) | 51–60+ | >20+ | online course, bedside teaching and didactics in residency | 51–100 |
| VAR002 | female | EM | none, EMS Assistant Director at Duke | 30–40 | 11–20 | ultrasound course or conference, rotation or elective (1 month), bedside teaching and didactics in residency | 101–150 |
| VAR003 | female | EM | none, Health Research Scholar | 30–40 | 0–10 | bedside teaching and didactics in residency | 51–100 |
| VAR004 | male | EM | EM Research director at Duke | 41–50 | 11–20 | bedside teaching and didactics in residency, informal sessions at academic institution/CME courses | >150 |
| VAR005 | female | EM | none | 51–60+ | >20+ | US course or conference (1–3 days) | 0–50 |
| VAR006 | male | EM | none, previous ED chief | 51–60+ | >20+ | US course or conference (1–3 days) | 0–50 |
| VAR007 | female | EM | none | 30–40 | 0–10 | rotation or elective, bedside teaching and didactics in residency and medical school | >150 |
| VAR008 | male | EM | Resident Education, station lead at US | 30–40 | 11–20 | ultrasound course or conference (1–3 days), rotation or elective, bedside teaching and didactics in residency | >150 |
| VAR009 | female | EM | Resident Education, station lead at US | 30–40 | 0–10 | ultrasound course or conference, rotation or elective (1 month), bedside teaching and didactics in residency | >150 |
| VAR010 | female | EM | ED chief | 41–50 | 11–20 | online course, bedside teaching and didactics in residency | 51–100 |
| VAR011 | female | EM | ED administration (Geriatric ED, sepsis committee, etc.) | 30–40 | 0–10 | rotation or elective, bedside teaching and didactics in residency and medical school | >150 |
| VAR012 | male | IM | Telehealth and IM resident orientation lead | 41–50 | 11–20 | none | 0–50 |
| VAR013 | male | IM | none | 41–50 | 11–20 | none | 0–50 |
| VAR014 | male | EM/APP | APP | 51–60+ | >20+ | none | 0–50 |

Key: EM = Emergency Medicine, IM = Internal Medicine, APP = Advanced Practice Provider, ED = emergency department, VA = Veterans Affairs, US = ultrasound, EMS = Emergency Medical Services.

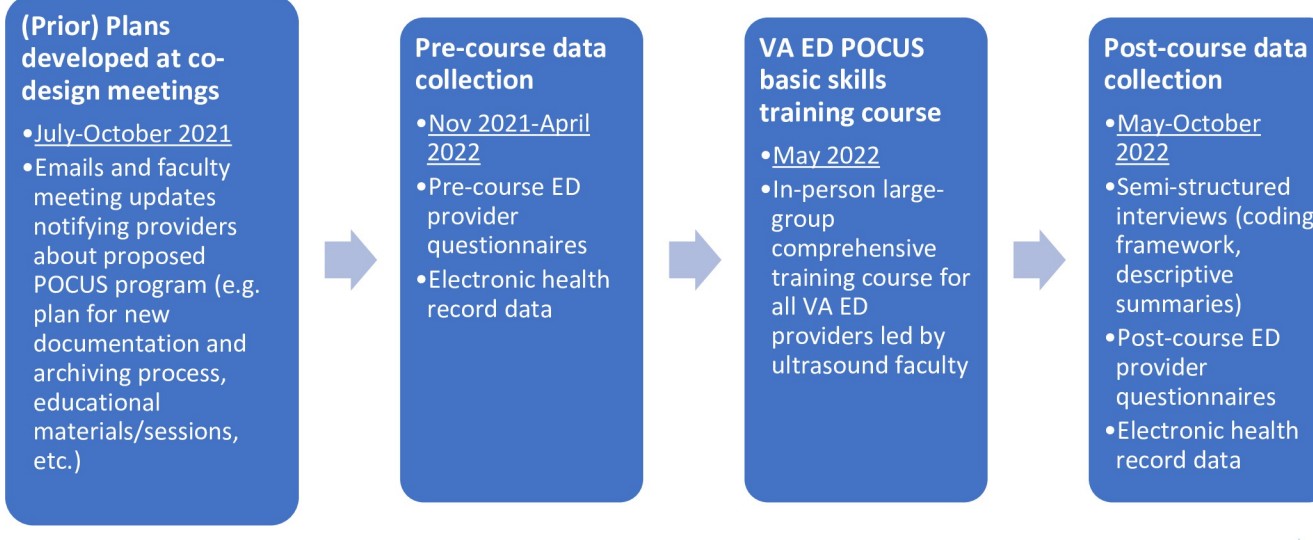

**Fig 2. Final proposed VA POCUS implementation intervention timeline.**

Summary of the five major themes and subthemes from ED provider interviews:

**Theme 1: POCUS workflow convenience and efficiency.** Participants desired a convenient and efficient clinical POCUS system to facilitate use within their ED clinical workflow. On the individual level, time constraints were a significant barrier to POCUS use. To facilitate POCUS use, participants preferred a straightforward and easy documentation system. They believed the fewer steps required for locating the ultrasound machines, image acquisition, documentation, and archiving, the better.

VAR002: *"The documentation can be so cumbersome. . .most important thing is. . .having a streamlined way of saving and documenting the images and being able to get trained on it so that you become proficient."*

Per participants, additional inner setting POCUS facilitators include increased clinical efficiency through resident teaching opportunities, safer procedures using ultrasound-guidance, and faster ED nursing intravenous (IV) access. For the inner/outer setting, participants commented on a hospital-wide POCUS archival system, including *"if they [the consultants] could just pull up the images to review themselves"* VAR007 and weekly image review by ultrasound-trained faculty to create *"a positive feedback loop"* VAR010, would facilitate POCUS use for continuity of patient care, medico-legal documentation purposes, and faculty credentialing in POCUS.

**Table 2. Identified themes and subthemes of POCUS barriers and facilitators as stratified by the Consolidated Framework for Implementation Research (CFIR).**

**CFIR construct: Individuals (number of participants/total participants)**

Themes: Convenience and efficiency, perceptions of high clinical utility of POCUS, perceptions of high educational utility of POCUS

- **Barriers:**
  - Time constraints based on clinical volume (14/14)
  - Comfort level with background POCUS knowledge and skills (14/14)
  - Eliminating unnecessary and cumbersome steps for image acquisition and documentation/storage (11/14)
- **Facilitators:**
  - Excellent image quality and ease of US machine use (14/14)
  - Producing differentials and diagnoses; answering pointed clinical questions or progression of specific diseases (11/14)
  - Provider clinical improvements and useful POCUS exams (10/14)
  - Ensuring patient safety (7/14)
  - Clinical standard of care (10/14)
  - Patient-centered care improvements with increased efficiency and satisfaction (8/14)
  - Hands-on POCUS group training rather than didactic learning (14/14)
- **Both:**
  - US machine usability and familiarity (14/14)
  - Ease of documentation (Barrier 4/14; Facilitator 8/14)
  - Desire for more practice and repetition (13/14)
  - Individuals' comfort with POCUS image acquisition and interpretation (Barrier 8/14; Facilitator 5/14)
  - Different teaching and learning styles for POCUS education: preferred hands-on, small group training over classroom didactics (14/14); human models (5/14); simulation (6/14)

**CFIR construct: Inner setting (Themes: ED environment and peer influences/feedback/teaching)**

- **Barriers:**
  - Availability of alternative radiology imaging modalities (11/14)
  - Location and cleanliness at other sites (but not a problem at the Durham VA) (12/14)
- **Facilitators:**
  - Availability of US machines and resources, cleanliness, and maintenance (14/14)
  - US learners and resident teaching (14/14)
  - POCUS use for ED procedures (14/14)
  - Existence of a POCUS archiving system and faculty credentialing system (12/14; 14/14)
  - POCUS quality assurance and feedback system (14/14)
  - Contagiousness and enthusiasm among colleagues (14/14)
  - Ultrasound faculty support and guidance (14/14)
- **Both:**
  - Necessity and utility of using ultrasound based on the clinical site's population (13/14)

**CFIR construct: Outer setting (VA hospital, other departments, and systemwide influences)**

- **Barriers:**
  - Theoretical implementation of a National VA policy with additional required steps (12/14)
- **Facilitators:**
  - Continuity of care with shared POCUS image viewing across the hospital system (2/14)
  - ED leadership and VA hospital leadership support (14/14)
  - Hospital-wide POCUS collaboration (14/14)
- **Both:**
  - Existence of a faculty credentialing system (Barrier 5/14; Facilitator 9/14)

Finally, participants perceived a National VA POCUS policy as a barrier, anticipating additional required steps to be time consuming, burdensome, and hindering clinical care based on experience with prior VA initiatives including procedural consents and moderate sedation (outer setting).

**Theme 2: ED environment and available resources (space and place).** Clean, readily available, up-to-date, and easy to locate ultrasound machines with high-quality image software facilitates POCUS use per participants. They report the VA's two diagnostic and one vascular cart-based ultrasound machines have intuitive buttons, a large screen for learners, excellent image quality, educational videos, and customizable exam templates to facilitate individuals' use. VAR006 *states "Here [at the VA], it's generally not hard to find them [ultrasound machines]. They are usually charged up and ready to go. . .If you know what you're doing, they're good images."* This contrasts to *"at [another local institution, name redacted] they're spread all over the place and it's hard to find them"* per VAR006, where participants report significant issues with antiquated machines, difficulties in locating a machine, and problems with network connectivity causing image upload failures as significant POCUS barriers.

**Theme 3: Perception of high clinical utility of POCUS in the ED (medical decision making).** Per participants, POCUS provides faster clinical data to guide patient management, especially in sicker patients that cannot be transported to radiology. For individuals, POCUS incorporation into their clinical workflow as an important diagnostic tool and for medical decision making facilitates use. Furthermore, participants report that POCUS is standard of care and that procedural ultrasound proficiency is a residency training requirement, which facilitates use. Several argue that ultrasound is more important than a stethoscope, calling it the *"stethoscope of the 21st century"*. Per VAR004 *". . .common complaints such as chest pain, abdominal pain, [and] shortness of breath where POCUS can help us eliminate some diagnoses or consider others. . .And of course using it when doing procedures like placing central lines or draining abscesses, that is where POCUS can be particularly helpful."*

Participants report POCUS improves patient safety and efficiency by reducing radiology travel and wait times (usually several hours). *"Also, patient satisfaction increases because they feel that you are doing more for them when you are bedside doing the ultrasound"* per VAR009.

They cite POCUS availability as crucial during nights and weekends when ultrasound technicians are off-site or unavailable, causing significant delays in patient care and poor satisfaction. Thus, they believe clinical POCUS use facilitates improved ED/hospital metrics and better patient care. *"I think that POCUS can be very beneficial. . .at night the ultrasound tech is not in the hospital . . .so the patient would have to sit in the ER until the morning, make an appointment for an ultrasound later, or you have to call in the tech in the middle of the night. Whereas if you had POCUS, you could just do it yourself. . .It saves time, I think it is a patient safety issue as well. . .why give a patient anticoagulation [for a possible DVT] when there is a way for us to safely do the ultrasound and properly document it?"* per VAR002.

**Theme 4: Perception of high educational utility of POCUS.** Participants overwhelmingly desired repetition and hands-on training to facilitate POCUS use. Per VAR009 *"More repetition plus training/doing/training. . .looking at pathology vs normal and spending enough time on this helps, especially the repetition."*

Those participants with minimal or no residency training (IM, FM, APP) desired basic training sessions on machine operation, knobology, probe selection/use, and image saving on ED-specific machines. Others requested advanced ultrasound training such as cardiac, testicular, ocular, biliary, DVT, and nerve blocks to improve their clinical practice.

Participants reported variable individual confidence in POCUS skills. Those providers practicing for 15–20 years without residency ultrasound training (IM/FM physicians and APP) perceived their lack of POCUS education as a barrier to use and did not feel comfortable

performing, interpreting, or teaching ultrasound. Most providers felt more comfortable with a subset of frequently performed exams, whereas advanced users with POCUS training in residency (e.g. younger EM graduates) reported comfort with more diagnostic and procedural exams, which facilitated use. VAR013 (IM physician) stated *"Again, I'm still at the level of which probe do I use. Even something like the FAST exam because I'm not EM. . . when I'm still trying to learn the first things, it's hard to retain."*

Participants report strong support from ED ultrasound faculty as a POCUS facilitator through clinical and monthly educational hands-on training sessions. They desired maximizing hands-on, small group teaching sessions, focusing on human models for advanced users and simulators for beginners to learn basic scanning techniques, probe positioning, and normal anatomy for skill development and retention.

Per VAR001 *"What would be helpful would be to have somebody come to the department while people are working and help the attendings and residents do the ultrasound on patients so that they could get teaching and hands-on training on patients at the same time."*

**Theme 5: Peer influences, feedback, and teaching.** For individuals, participants reported peer enthusiasm motivated POCUS integration into their practice, citing POCUS teaching interventions and image feedback as additional facilitators. Participants felt contagious enthusiasm in POCUS use. *"Yes, it's nice to see enthusiasm from other people for teaching, and when people are sharing the images, it gets other people to also try, [which] could make ultrasound more mainstreamed"* per VAR009. Participants perceived ultrasound faculty as approachable on shift, which they felt increased providers' likelihood to use POCUS. Participants also felt reassured witnessing others' strengths and weaknesses while learning together in a group environment. They perceived that peer influence improves group knowledge and creates positive feedback: *"It's infectious when someone says, 'oh look how awesome this is, let me show you how to do it'. Or when they say 'the resident and I saw this cool thing', you know it's like peer pressure. It's how you change culture. . .if everyone else around me is doing it, it would make me want to as well"* per VAR010, which facilitates sustained use and retention in the inner setting.

For POCUS teaching and future educational interventions, participants noted ED ultrasound faculty's enthusiasm, facilitating POCUS use for individuals and in the inner setting. VAR004 shared **"***Well, I liked that it was individualized. We did the exams that people wanted to learn and study the most. Also, it was hands on, so we got to actually do the exams."*

Their comprehensive POCUS training course for ED providers with rotating hands-on scanning sessions and mixed simulation and human models maximized learning. Longitudinal learning with monthly hands-on repetition and 5-minute topics during faculty meeting were preferred learning methods and were perceived to facilitate sustained POCUS use and knowledge/skill retention per participants. Per VAR008 *"Continued education is probably key and there's like I said more advances in ultrasound- new nerve blocks and stuff like that–staying up to date on that is the tricky part."*

For the outer setting, participants believed *"anytime that we collaborate with other departments it builds better communication between the departments and strengthens relationships"* VAR001. Specialists (e.g. hospital medicine, intensivists) and ED ultrasound faculty could teach each other tips, different techniques, or methods, facilitating POCUS use.

Finally, participants perceived leadership support as key to program success and departmental growth. They believed that ED leadership had supported ultrasound faculty teaching time and resources and were active participants in the training sessions, facilitating POCUS use through example and peer influences (inner setting). *"They [ED leadership] are trying to incorporate a lot more with [ultrasound faculty]. I think they are pretty supportive."* VAR007 reported.

Regarding VA hospital leadership (outer setting), participant VAR006 stated *"I'm not really sure where they stand. They're not necessarily a hindrance, I don't really think they know what we've been doing in the ED."* Participants perceived a lack of national or regional POCUS initiatives, as hospital leadership purchased our updated ultrasound machines but failed to establish an image archiving system over the past year.

The codebook with major themes/subthemes is summarized in the supplement (S1 Table). Further descriptive data including individual interview quotes are included in the supplement (S2 Table).

## 4 Discussion

Applying our findings to the CFIR allows us to identify priority areas and key stakeholders to increase identified facilitators and reduce barriers after POCUS program implementation [19]. Through semi-structured interviews, participants have identified a straightforward, clinical flow process as an essential facilitator to POCUS use, with the implementation of a standardized documentation, archival, and image review process as additional key facilitators. Barriers to POCUS training and education remain and can be overcome through longitudinal, repetitive hands-on teaching sessions by US-skilled faculty. Finally, using momentum from peer enthusiasm, collaboration, and positive feedback can help propel POCUS efforts in achieving sustainability over time.

First, motivation from the individual provider remains key in POCUS education, knowledge, and utilization. The comprehensive POCUS training session was very well received per participants' input and if compared to a professional comprehensive ultrasound course would cost >$1,000 per participant (or >$15,000 total). Training of this caliber requires faculty time, high-level US resources, schedule coordination, and collaboration for success [1,7]. Also, many providers preferred brief ultrasound tips via email and at monthly faculty meetings, citing minimal available non-clinical time for learning. Few participants expressed interest in classroom didactics, which is consistent with curriculum changes from classroom-style presentations to active learning with hands-on skill acquisition [31]. Participants believed that tailoring educational interventions to the individual ED providers and group's needs would facilitate POCUS use, with repetitive sessions over a longitudinal timeline for reinforcement.

Participants strongly supported POCUS's high educational value, perceiving resident and student learners as facilitators to POCUS use as ultrasound is standard of care for both diagnostic and procedural ED applications and is being incorporated in residency training programs [2,3]. POCUS facilitates both the teachers' and learners' sustained use in clinical care and for knowledge retention, as identified in other POCUS studies involving learners [15,21].

Regarding high perceived clinical utility of POCUS, participants believed patient complexity and medical presentation highly affects POCUS use. This was evident in our POCUS findings, with the most common POCUS examinations being performed for cardiac then abdominal presentations, patients with shortness of breath to evaluate for pulmonary edema or effusions, and finally to evaluate volume status. These matched the most common ED presentations per our VA site's demographics. Furthermore, participants felt the lack of radiology US technician availability at night and on weekends was a facilitator in promoting POCUS use, with high potential to improve patient care and ED metrics including disposition times for patients undergoing radiology US studies, especially regarding deep vein thrombosis (DVT) and biliary ultrasound at our VA site (~41% of total ED radiology ultrasounds were for DVT and ~10% for biliary). Procedural POCUS facilitated use by increasing efficiency and patient safety, with participants citing use for soft tissue (evaluating abscess vs cellulitis), MSK applications (joint effusion aspiration), paracentesis, and IV access including central lines.

For the underline{inner setting}, like prior self-reported survey-based studies, time constraints and disruptions to clinical efficiency and flow were large barriers to POCUS use [11,13]. Anything that increased the number of steps required to perform an ultrasound or slowed down clinical efficiency was perceived as a barrier [15]. Based on user feedback, we identified a need for better communication and information dissemination methods for a simplified clinical POCUS workflow process. Furthermore, some providers reported not performing or saving POCUS images and instead ordering radiology US studies given concerns about medico-legal documentation purposes for their clinical encounter, which is also a deterrent at other sites [9,10]. A standardized image documentation and archival system across the VA system would alleviate this problem.

Fortunately, per participants at this VA ED, ultrasound machine location, cleanliness, and functionality were not a problem. They report excellent image quality and a user-friendly, intuitive interface, which facilitates ultrasound use. This contrasts to self-reported survey data citing problems with lack of updated ultrasound machines and poor POCUS program infrastructure as significant barriers to POCUS use at other sites [9–13]. Difficulties remain in enforcing a standardized system for saving patient images and performing appropriately documented POCUS (e.g. preventing incomplete studies or the habit of not saving images or "phantom scans"), which is a common problem in EDs across the country [15,21]. Development of an image archival system and image review process by ultrasound-trained faculty were perceived as facilitators to POCUS use, which is a significant need identified in VA studies and should be a focus of future POCUS implementation endeavors on the regional and national VA level [12,13,25].

For the underline{outer setting}, hospital sites with buy-in from ED and hospital leadership likely may have higher rates of success and sustainability than those that lack support [32]. Specifically, the involvement of ED leadership with co-designing our training intervention from the start, as well as monthly in-person reinforcement sessions, allowed us to reach a greater number of individuals including attending physicians, APPs, residents, and students [18,33,34]. Participants agreed that collaboration between physician and nursing staff, as well as between departments, further strengthened the use of POCUS and its influence on patient care. Moreover, by creating a positive and cohesive environment within the ED with collegial enthusiasm and positive feedback on shift, we were able to increase awareness and achieved a four-fold increase in clinical POCUS use within 12 months [16].

Interestingly, providers reported that a national VA policy would hinder POCUS use if it became too cumbersome and created additional steps in image documentation, as occurred with other processes such as moderate sedation, procedural consents, opioid prescribing, and medication orders in the VA system. They also thought a faculty credentialing system would be a barrier if it created additional required steps but could be a facilitator with proper system integration.

## Limitations

Our clinical site currently has only one EM ultrasound fellowship-trained faculty and several other regular ultrasound users, thus our teaching and dissemination efforts are limited by faculty time and availability (although this is likely realistic of other VA or community sites nationwide). In addition, most VA hospitals, including our site, do not yet have a POCUS archiving system in place, which significantly limits the ability to perform image review, faculty credentialing, and closed-loop feedback to providers. Thus, increased local and national funding for POCUS faculty support and institutional infrastructure are essential in future POCUS program implementation. Another limitation is that our VA site is strongly affiliated

with an academic center (4/14 participants also work at the academic center), thus the participants may be more enthusiastic about POCUS participation than the general VA workflow, although many VA sites are affiliated with an academic institution per prior studies [12,13,25].

### Future studies

More work is needed to reinforce and sustain POCUS knowledge and skills amongst a diverse group of clinical providers. Each clinical site should tailor POCUS education and clinical use toward the needs of the individual providers and the ED group. VA sites with an ED "clinical champion" for a particular endeavor have performed well in other research studies, and this could be applied to POCUS as well [7,32]. Collaboration between physician and nursing groups can be further explored, particularly with overlapping roles such as performance of US-PIV placement and bladder volume scans, which are common ED procedures. Future studies can use our model to evaluate potentially different implementation strategies based on unique barriers or facilitators at each site.

### Conclusions

Our qualitative study describes a reproducible method of site evaluation to identify facilitators and barriers after the implementation of a co-designed POCUS training intervention at a single VA ED. Future work can use this model as a clinical tool to rapidly inform an intervention strategy through barrier and facilitator identification at each site.

### Supporting information

**S1 Appendix. Semi-structured interview questions on point-of-care ultrasound barriers and facilitators at a single site emergency department.**
(DOCX)

**S1 Table. Veterans Affairs point-of-care ultrasound implementation study codebook.**
(DOCX)

**S2 Table. Identified themes and codes and respective barriers and facilitators as stratified by the Consolidated Framework for Implementation Research (CFIR).**
(XLSX)

### Author Contributions

**Conceptualization:** Rebecca G. Theophanous, Luna Ragsdale, Catherine A. Staton, Stephanie A. Eucker.

**Data curation:** Rebecca G. Theophanous, Padmaja Krishnan, Raelynn Vigue, Carson Herman, Jaran White.

**Formal analysis:** Rebecca G. Theophanous, Anna Tupetz, Padmaja Krishnan, Raelynn Vigue, Carson Herman, Jaran White, Stephanie A. Eucker.

**Funding acquisition:** Rebecca G. Theophanous, Luna Ragsdale, Catherine A. Staton.

**Investigation:** Rebecca G. Theophanous, Anna Tupetz, Luna Ragsdale.

**Methodology:** Rebecca G. Theophanous, Anna Tupetz, Padmaja Krishnan, Catherine A. Staton, Stephanie A. Eucker.

**Project administration:** Luna Ragsdale, Padmaja Krishnan, Raelynn Vigue, Carson Herman, Jaran White.

**Resources:** Luna Ragsdale, Catherine A. Staton, Stephanie A. Eucker.

**Software:** Anna Tupetz.

**Supervision:** Rebecca G. Theophanous, Anna Tupetz, Luna Ragsdale, Catherine A. Staton, Stephanie A. Eucker.

**Writing – original draft:** Rebecca G. Theophanous.

**Writing – review & editing:** Rebecca G. Theophanous, Anna Tupetz, Luna Ragsdale, Padmaja Krishnan, Raelynn Vigue, Carson Herman, Jaran White, Catherine A. Staton, Stephanie A. Eucker.

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
