## [Decision Letter · Decision Letter 0]

7 Aug 2024

PONE-D-24-03510A qualitative study of perceived barriers and facilitators to point-of-care ultrasound use among Veterans Affairs Emergency Department providersPLOS ONE

Dear Dr. Theophanous,

Thank you for submitting your manuscript to PLOS ONE. After careful consideration, we feel that it has merit but does not fully meet PLOS ONE’s publication criteria as it currently stands. Therefore, we invite you to submit a revised version of the manuscript that addresses the points raised during the review process.

Please address a minor concern described below.

We look forward to receiving your revised manuscript.

Kind regards,

Takeru Abe, Ph.D

Academic Editor

PLOS ONE

2. In the ethics statement in the Methods, you have specified that verbal consent was obtained. Please provide additional details regarding how this consent was documented and witnessed, and state whether this was approved by the IRB

“RGT received a Society of Academic Emergency Medicine Foundation/Academic Emergency Ultrasound Section grant #AG 2020-0000000099. https://www.saem.org/about-saem/academies-interest-groups-affiliates2/saem-foundation/apply-for-a-grant/academy-grants/aeus-research-grant.”

“This study was funded by an SAEMF-AEUS grant #AG 2020-0000000099.”

“RGT received a Society of Academic Emergency Medicine Foundation/Academic Emergency Ultrasound Section grant #AG 2020-0000000099. https://www.saem.org/about-saem/academies-interest-groups-affiliates2/saem-foundation/apply-for-a-grant/academy-grants/aeus-research-grant.”

Reviewers' comments:

Reviewer's Responses to Questions

**Comments to the Author**

1. Is the manuscript technically sound, and do the data support the conclusions?

Reviewer #1: Yes

2. Has the statistical analysis been performed appropriately and rigorously? 

Reviewer #1: N/A

3. Have the authors made all data underlying the findings in their manuscript fully available?

Reviewer #1: Yes

4. Is the manuscript presented in an intelligible fashion and written in standard English?

Reviewer #1: Yes

5. Review Comments to the Author

Reviewer #1: Thank you for giving an opportunity to review this interesting work.

The authors described a qualitative study regarding barriers and facilitators to POCUS at EM department of VA Hospital. They presented data with sound methodologies.

This reviewer has a minor suggestion; it would be fruitful for readers if the authors add a description and an explanation regarding with analyses they adopted and did not adopt, such as content analysis with grounded theory and thematic analysis.

This information could help readers to understand your methodology better, so that the readers will be able to judge the findings' generalizability.

6. PLOS authors have the option to publish the peer review history of their article (what does this mean?). If published, this will include your full peer review and any attached files.

Reviewer #1: No

---

## [Author Response · Author response to Decision Letter 0]

28 Aug 2024

-The manuscript and title page were reformatted per journal requirements.

2. In the ethics statement in the Methods, you have specified that verbal consent was obtained. Please provide additional details regarding how this consent was documented and witnessed, and state whether this was approved by the IRB. – Added two sentences to the methods section: “Our study was deemed exempt by the VA Institutional Review Board, and consent was waived by the IRB (1631300-4).” “Verbal consent was obtained by reading aloud a written statement describing voluntary study participation at the beginning of each audio recorded interview and documented using a data collection checklist.”

“RGT received a Society of Academic Emergency Medicine Foundation/Academic Emergency Ultrasound Section grant #AG 2020-0000000099. https://www.saem.org/about-saem/academies-interest-groups-affiliates2/saem-foundation/apply-for-a-grant/academy-grants/aeus-research-grant.”

Please state what role the funders took in the study. If the funders had no role, please state: "The funders had no role in study design, data collection and analysis, decision to publish, or preparation of the manuscript." – This statement was added to the cover letter.

“This study was funded by an SAEMF-AEUS grant #AG 2020-0000000099.”

“RGT received a Society of Academic Emergency Medicine Foundation/Academic Emergency Ultrasound Section grant #AG 2020-0000000099. https://www.saem.org/about-saem/academies-interest-groups-affiliates2/saem-foundation/apply-for-a-grant/academy-grants/aeus-research-grant.” – The funding statement is correct. The text was removed from the manuscript and added to the cover letter.

5. Please review your reference list to ensure that it is complete and correct. If you have cited papers that have been retracted, please include the rationale for doing so in the manuscript text, or remove these references and replace them with relevant current references. Any changes to the reference list should be mentioned in the rebuttal letter that accompanies your revised manuscript. If you need to cite a retracted article, indicate the article’s retracted status in the References list and also include a citation and full reference for the retraction notice. – References were reviewed and outdated website links updated.

Reviewers' comments:

Reviewer's Responses to Questions

Comments to the Author

1. Is the manuscript technically sound, and do the data support the conclusions?

Reviewer #1: Yes

2. Has the statistical analysis been performed appropriately and rigorously?

Reviewer #1: N/A

3. Have the authors made all data underlying the findings in their manuscript fully available?

Reviewer #1: Yes

4. Is the manuscript presented in an intelligible fashion and written in standard English?

Reviewer #1: Yes

5. Review Comments to the Author

Reviewer #1: Thank you for giving an opportunity to review this interesting work.

The authors described a qualitative study regarding barriers and facilitators to POCUS at EM department of VA Hospital. They presented data with sound methodologies.

This reviewer has a minor suggestion; it would be fruitful for readers if the authors add a description and an explanation regarding which analyses they adopted and did not adopt, such as content analysis with grounded theory and thematic analysis.

This information could help readers to understand your methodology better, so that the readers will be able to judge the findings' generalizability. -Details were added to the methods section about the analysis methodologies used as suggested by the reviewers.

6. PLOS authors have the option to publish the peer review history of their article (what does this mean?). If published, this will include your full peer review and any attached files.

Do you want your identity to be public for this peer review? For information about this choice, including consent withdrawal, please see our Privacy Policy.

Reviewer #1: No

---

## [Editor Report · Decision Letter 1]

2 Sep 2024

A qualitative study of perceived barriers and facilitators to point-of-care ultrasound use among Veterans Affairs Emergency Department providers

PONE-D-24-03510R1

Dear Dr. Theophanous,

We’re pleased to inform you that your manuscript has been judged scientifically suitable for publication and will be formally accepted for publication once it meets all outstanding technical requirements.

Kind regards,

Takeru Abe, Ph.D

Academic Editor

PLOS ONE
---

## [Editor Report · Acceptance letter]

4 Sep 2024

PONE-D-24-03510R1 

PLOS ONE

Dear Dr. Theophanous, 

I'm pleased to inform you that your manuscript has been deemed suitable for publication in PLOS ONE. Congratulations! Your manuscript is now being handed over to our production team.

Kind regards, 

on behalf of

Dr. Takeru Abe 

Academic Editor

PLOS ONE